# Analysis of Shear Model for Steel-Fiber-Reinforced High-Strength Concrete Corbels with Welded-Anchorage Longitudinal Reinforcement

**DOI:** 10.3390/ma16144907

**Published:** 2023-07-09

**Authors:** Shu-Shan Li, Die Peng, Heng Wang, Feng-Jian Zhang, Hong-Mei Li, Yi-Jun Xie, Ai-Jiu Chen, Wei Xie

**Affiliations:** 1School of Civil Engineering and Communication, North China University of Water Resources and Electric Power, Zhengzhou 450046, China; lishushan@ncwu.edu.cn (S.-S.L.); caj@ncwu.edu.cn (A.-J.C.); xwei@ncwu.edu.cn (W.X.); 2Engineering Technology Research Center for Structural Vibration Control and Health Monitoring of Henan Province, Zhengzhou 450046, China; 3Powerchina Huadong Engineering Corporation Limited, Hangzhou 311122, China; wang_h27@hdec.com; 4School of Civil and Transportation Engineering, Henan University of Urban Construction, Pingdingshan 467036, China; 30010608@hncj.edu.cn; 5Powerchina Road Bridge Group Co., Ltd., Beijing 100160, China; xieyijun87@163.com

**Keywords:** corbel, steel-fiber-reinforced high-strength concrete (SFHSC), welded-anchoragelongitudinal reinforcement, shear span ratio, steel fiber volume fraction, shear transfer mechanism, modified softened strut-and-tie model (MSSTM)

## Abstract

According to the shear capacity test results of six steel-fiber-reinforced high-strength concrete (SFHSC) corbels with welded-anchorage longitudinal reinforcement under concentrated load, the effects of shear span ratio and steel fiber volume fraction on the failure mode, cracking load and ultimate load of corbel specimens were analyzed. On the basis of experimental research, the shear transfer mechanism of corbel structure was discussed. Then, a modified softened strut-and-tie model (MSSTM), composed of the diagonal and horizontal mechanisms, was proposed, for steel-fiber-reinforced high-strength concrete corbels. The contributions of concrete, steel fiber and horizontal stirrups to the shear bearing capacity of the corbels were clarified. A calculation method for the shear bearing capacity of steel-fiber-reinforced high-strength concrete corbels was established and was simplified on this basis. The calculation results of the model were compared with the test values and calculation results of the GB50010-2010 code, the ACI318-19 code, the EN 1992-1-1 code and the CSA A23.3-19 code. The results showed that the concrete corbel with small shear span ratio mainly has two typical failure modes: shear failure and diagonal compression failure. With the increase in shear span ratio, the shear capacity of corbels decreases. Steel fiber can improve the ductility of a reinforced concrete corbel, but has little effect on the failure mode of the diagonal section. The calculated values of the national codes were lower than the experimental values, and the results were conservative. The theoretical calculation values of the shear capacity calculation model of the corbels were close to the experimental results. In addition, the model has a clear mechanical concept considering the tensile properties of steel-fiber-reinforced high-strength concrete and the influence of horizontal stirrups, which can reasonably reflect the shear transfer mechanism of corbels.

## 1. Introduction

Corbels are defined as short cantilevers extending from a wall or column, usually used to support other superstructures such as prefabricated beams or floor slabs at building joints. They are important vertical local-compression components in structures. Due to the influence of concrete creep, shrinkage, thermal deformation and earthquake, corbels also endure certain horizontal loads [1]. When applied to cast-in-place and prefabricated structures, the requirements for the crack resistance and bearing capacity of corbels are high. Steel-fiber-reinforced high-strength concrete (SFHSC) is a new type of composite material formed by adding steel fibers that can enhance, toughen and resist cracking in concrete. The addition of steel fibers significantly improves the tensile, flexural and shear strength of concrete. Adding steel fibers to reinforced concrete corbels can effectively improve the load-bearing capacity and seismic performance of the corbels and improve the degree of reinforcement density [2]. Due to the presence of geometric dimensions and concentrated loads, the corbels are classified as discontinuous regions (D-regions), exhibiting nonlinear strain distribution under load, which does not conform to the plane section assumption of bending theory. The strength of corbels is mainly controlled by shear rather than bending [3], and the shear mechanism is complex, similar to the variable cross-section deep beam. Therefore, the traditional cross-sectional design method based on the plane section assumption is no longer applicable to such components.

Many scholars have systematically studied the mechanical properties of concrete corbels. Fattuhi [4,5] conducted vertical loading tests on reinforced concrete corbels using steel fibers as shear-toughening materials. The tests showed that the addition of steel fibers improved the ductility and strength of the corbels. When the longitudinal reinforcement ratio in the corbels was low, the failure mode changed from diagonal splitting or shear to bending, and the bending model method and truss model method were used to predict the bearing capacity of corbels. Campione et al. [6] studied the flexural behavior of corbels, and the results showed that adding fibers and configuring stirrups were beneficial to improve the bearing capacity and ductility of concrete corbels. A simplified calculation model of shear strength for fiber-reinforced concrete corbels with stirrups was proposed. Yang et al. [7] conducted experimental research on concrete corbels with different steel fiber volume fraction, which used two anchorage methods for the main tension tie: headed bars and welded anchorage. It was concluded that the corbels with headed bars have higher bearing capacity, stiffness and ductility. Gao et al. [8] discussed the mechanical properties and failure characteristics of steel-fiber-reinforced high-strength concrete. The study showed that the addition of steel fibers increased the diagonal crack load and ultimate load of the corbels, and the brittleness was improved. Gao et al. [9] concluded through the shear test of steel-fiber-reinforced concrete corbels that steel fiber has little effect on the failure mode of the corbels, but it can improve the ductility of the corbels. The shear bearing capacity of the corbel increases with the increase in steel fiber volume fraction and concrete strength, and it decreases with the increase in shear span ratio. At the same time, a formula for calculating the shear bearing capacity of steel-fiber-reinforced concrete corbels was derived. Kurtoglu et al. [10] proposed an empirical formula based on symbolic regression (SR) to predict the ultimate shear strength of steel-fiber-reinforced concrete and glass-fiber-reinforced concrete corbels without stirrups and verified its applicability. Huang et al. [11] analyzed the calculation method for the shear bearing capacity of the corbel based on the vertical shear test database of corbels, and they fitted a simplified formula for the design correction coefficient of the shear bearing capacity of corbels according to the parameter analysis results. The Chinese code GB50010-2010 [12] uses the triangular truss model for force analysis of corbels, but the proposed shear capacity formula is a semi-theoretical and semi-empirical formula based on the test, lacking a clear mechanical concept. Foreign scholars put forward the strut-and-tie model (STM) by improving the early truss model, making it a universal design method. The American code ACI318-19 [13], the European code EN 1992-1-1 [14] and the Canadian code CSA A23.3-19 [15] all introduce the strut-and-tie model for the bearing capacity design of D-region components. The strut-and-tie model has been widely used and continuously improved at home and abroad [16,17,18,19,20]. Subsequently, Hwang and Lee proposed the softened strut-and-tie model (SSTM) based on the strut-and-tie model, which considered the softening characteristics of concrete under compression, and its applicability was tested in shear analysis of various concrete components [21,22,23,24]. Khosravikia et al. [25] carried out finite element numerical simulation on the corbel structure, analyzed the shear transfer mechanism of the corbel structure and the influencing factors of the concrete strut coefficient, and improved the calculation method for the shear bearing capacity of corbels based on the strut-and-tie model. Based on the finite element method, Canha et al. [26] conducted research and analysis on parameters such as shear span ratio and reinforcement ratio using validated models, and they compared them with experimental results. The numerical simulation results were good.

Scholars have carried out a considerable amount of research on the SFHSC corbel, and put forward many calculation methods for the SFHSC corbel, but its applicability remains to be further verified. The current specifications for a SFHSC corbel design method are not perfect. In this study, experimental research and theoretical analysis on the shear performance of six SFHSC corbels with welded-anchorage longitudinal reinforcement were carried out. The effects of shear span ratio and steel fiber volume fraction on the failure mode, cracking load and ultimate load of corbel components were analyzed. According to the mechanical characteristics of steel-fiber-reinforced high-strength concrete corbels, the steel fibers distributed randomly in the concrete were equivalent to horizontal and vertical micro-reinforcement. Then, based on the softened strut-and-tie model, a calculation model for the shear bearing capacity of steel-fiber-reinforced high-strength concrete corbels was established. We calculated and compared the shear bearing capacity of the six steel-fiber-reinforced high-strength concrete corbels tested in this paper using the calculation model and national codes. The shear mechanism of corbels was analyzed. The applicability of the calculation model was verified, and the research related to the SFHSC corbel was further improved.

## 2. Experimental Study

### 2.1. Specimen Design and Production

Six SFHSC corbel specimens with welded-anchorage longitudinal reinforcement specimens, numbered CW01 to CW06, were designed and manufactured for the experiment. Two control variables were set: shear span ratio and steel fiber volume fraction. The first group, CW01 to CW04, had shear span ratios of 0.2, 0.3, 0.4 and 0.5, respectively; the second group, CW02, CW05 and CW06, had steel fiber volume fractions of 1.5%, 0.75% and 0%, respectively. The cross-sectional width of the corbels was 200 mm, and the cantilever height of the outer corbels was 200 mm, with a total height of 450 mm. The reinforcement amount for each specimen was the same. Four hot-rolled ribbed steel bars with a diameter of 12 mm were used for the longitudinal reinforcement, with a reinforcement ratio of 0.55%. Ten smooth, round steel bars with a diameter of 10 mm were used for the stirrups, with a reinforcement ratio of 0.78%. The longitudinal reinforcements were arranged by welding and anchoring. In order to facilitate the monitoring of steel bar strain during the later loading process, six steel bar strain gauges were arranged on the longitudinal tensile reinforcement 10 cm away from the corbel lower column, symmetrically arranged on both sides (one longitudinal reinforcement was not measured). The steel bar strain gauges on the corbel stirrup were also evenly distributed on both sides 10 cm away from the corbel lower column. The size, reinforcement and reinforcement strain gauge arrangement of the corbel specimens are shown in Figure 1, and the design parameters of the specimens are shown in Table 1.

C60 high-strength concrete was used in the experiment. The coarse aggregate was gravel with a particle size of no more than 20 mm. The fine aggregate was medium coarse sand. The gravel and sand were evenly graded, and the cement was 42.5 grade ordinary Portland cement. Milling steel fiber with a length of 32 mm, a diameter of 0.75 mm, a length-to-diameter ratio of 42.7 and a tensile strength greater than or equal to 600 MPa was added. The specific mix ratio of the SFHSC is shown in Table 2. Six 150 mm × 150 mm × 150 mm cube test blocks were reserved to measure the cube compressive strength and the tensile strength of the SFHSC. Six 150 mm × 150 mm × 300 mm prism test blocks were reserved to measure the axial compressive strength and the elastic modulus of the SFHSC. The measured mechanical properties of the SFHSC and the steel bars are shown in Table 3 and Table 4, respectively.

### 2.2. Measuring Points Arrangement and Loading Scheme

In order to conduct real-time observation on the strain changes of the concrete surface at the corbel normal section and the diagonal section during the test, and control the test process, 16 concrete strain gauges were arranged at the positions where the surface of each corbel specimen might be damaged, including 8 normal sections and 8 inclined sections (changing with the shear span ratio), symmetrically arranged on both sides, as shown in Figure 2. The corbel columns and supports were respectively set up with displacement meters to measure the deflection under each level of load and make settlement. The strain and displacement data of the steel and concrete were measured using an IMP data acquisition instrument. The crack width was observed using a KON-FK (N) crack observation instrument with a minimum scale of 0.01 mm.

The experimental loading device was a 500 t three-beam, four-column hydraulic press. The symmetrical loading method was adopted. The size of the loading plate was 400 mm × 80 mm × 20 mm. The contact mode between the loading point and the press was non-rigid constraint. A constraint in the horizontal direction was reduced through the circular shaft, and the loading mode was maintained as vertical loading. In order to prevent stress concentration, fine sand was evenly smeared between the loading mold and the component, as shown in Figure 3. The specimen was preloaded before loading, the gap was compacted to level the loading device and the instrument was tested for normal operation. The preload size was 30 kN. After the preload was completed, the test monitoring data were cleared. Then, the load was increased to 480 kN per 60 kN level and kept stationary for 2 min to stabilize the readings before recording. After 480 kN, each level was loaded with 100 kN until failure. After the specimen was destroyed, the data were recorded every 100 kN in the descending stage until the test was completed when it dropped to 800 kN.

### 2.3. Test Results

The characteristic loads and failure modes for each stage of the corbels are shown in Table 5. The load when the first vertical crack occurred in the specimen was defined as the normal section cracking load, denoted by $V_{cr}^{N}$. The load corresponding to the first diagonal crack in the specimen was the diagonal section cracking load, which was recorded as $V_{cr}^{D}$. $V_{u}$ was the ultimate load that the specimen could bear. It can be seen from Table 5 that the cracking load of the normal section of steel-fiber-reinforced high-strength concrete corbels was about 15% to 25% of the ultimate load, and the cracking load of the diagonal section was about 40% to 60% of the ultimate load. Two typical failure modes of shear failure and diagonal compression failure occurred. As the shear span ratio increased, the bearing capacity of the steel-fiber-reinforced high-strength concrete corbels decreased significantly, and the failure mode gradually transformed from shear failure to diagonal compression failure. When the shear span ratio was the same, with the addition of steel fibers, the bearing capacity of the corbel increased with the increase in steel fiber volume fraction.

### 2.4. Failure Process and Failure Forms

The failure form and crack development of each specimen are shown in Figure 4.

When the shear span ratio λ=0.2, the corbel undergoes shear failure. The typical specimen CW01 can be taken as an example to illustrate the shear failure process, and the failure mode is shown in Figure 5a. In this type of failure, the connection between the loading point and the junction point under the corbel column was the cracking connection, and the crack ran through the whole corbel component. When the load was about 20% to 30% of the failure load, the first crack appeared. At the same time, there were many small cracks under the loading plate, but the width of each crack was very small. When the load continued to 40% to 50% of the failure load, the crack continued to extend up and down from the midpoint of the crack, and the speed of crack width increase and extension speed increased together. Afterwards, as the load increased, the crack would eventually develop to the junction of the corbel and the column when the load was approaching the final failure. Finally, the cracks were connected up and down into a small diagonal crack and formed rapidly, converging into the widest crack. The specimen was cut along this crack and destroyed, at which point the longitudinal reinforcement did not reach its yield strength.

When the shear span ratio λ≥0.3, the corbel undergoes diagonal compression failure. The typical specimen CW03 can be taken as an example to illustrate the diagonal compression failure process, and the failure mode is shown in Figure 5b. When the load was increased to 17% to 28% of the ultimate load, the first crack would appear roughly. It had little effect on the mechanical properties of the corbels due to the newly generated crack width at this time. However, as the load continued to increase, the deflection of the component would have a certain increase, and the axis of the rod would also form a certain bending. In this case, when the load reached 40% to 60% of the failure load, the connecting line between the loading plate and the junction of the corbel lower part and the column, which was equivalent to the concrete strut, would generate a large number of cracks. After this, the load continued to increase. When the load was close to failure (about 80% of the failure load), a large number of new diagonal cracks suddenly appeared in the compression area, and the concrete skin cracked. Finally, the top tensile steel bar reached yield first, and the strain suddenly increased beyond the range. The concrete reached its axial compressive strength, and the specimen subsequently failed.

### 2.5. Analysis of Influencing Factors

#### 2.5.1. Shear Span Ratio

The effects of the shear span ratio on the cracking loads of the normal and diagonal sections and the ultimate load of the corbels were analyzed using specimens CW01 (0.2), CW02 (0.3), CW03 (0.4) and CW04 (0.5). It can be seen from Figure 6 that the shear span ratio is an important factor affecting the bearing capacity of the corbels. As the shear span ratio increased, the bearing capacity of the corbels decreased significantly, and the cracking loads of the normal and diagonal sections and the ultimate load decreased accordingly. For the normal section cracking load, CW02, CW03 and CW04 decreased by 9.5%, 32.4% and 42.9%, respectively, compared with CW01. For the diagonal section cracking load, CW02, CW03 and CW04 decreased by 6.7%, 27.6% and 32.6%, respectively, compared with CW01. For the ultimate load, CW02, CW03 and CW04 decreased by 15.6%, 22.6% and 24.0%, respectively, compared with CW01. The influence of the shear span ratio on the cracking load of the diagonal section of the corbels can be explained as follows: the main tensile stress will be generated in the vertical direction of connection between the concentrated load point and the lower column of the corbels, while the vertical compressive stress at the loading point and the bearing reaction force will reduce the main tensile stress, thereby improving the bearing capacity of the corbels. The cracking load of the diagonal section increases, and the vertical compressive stress increases with the decrease in the shear span ratio, and the cracking load of the diagonal section of the corbels increases.

#### 2.5.2. Steel Fiber Volume Fraction

Figure 7 shows the effect of the steel fiber volume fraction on the cracking loads of the normal and diagonal sections and the ultimate load of corbel specimens CW02 (1.5%), CW05 (0.75%) and CW06 (0%) with a shear span ratio of 0.3. It can be seen from the diagram that the cracking loads of the normal and diagonal sections and the ultimate load of the corbels increased with the increase in steel fiber volume fraction. For the normal section cracking load, CW02 increased by 26.7% compared to CW05 and 18.8% compared to CW06. For the diagonal section cracking load, CW02 increased by 2.6% compared to CW05 and 8.6% compared to CW06. For the ultimate load, CW02 increased by 12.8% compared to CW05 and 11.4% compared to CW06. 

### 2.6. Concrete Strain

In Figure 8, the typical specimen CW02 is taken as an example to analyze the concrete strain of the normal and diagonal sections of the steel-fiber high-strength concrete corbel under load. It can be seen from the figure that the concrete strain of the normal section of the corbel specimen was obviously more intense than that of the diagonal section. The further away from the loading point it was, the lower the concrete strain was for both the normal and diagonal sections, which was the result of the torque change. It can be seen from Figure 8a that, before the cracking of the steel-fiber-reinforced high-strength concrete corbel, the normal section strain was small and basically linear, which satisfied the plane section assumption. With an increase in load, the strain of the normal section concrete increased nonlinearly, which did not conform to the plane section assumption. This indicated that the addition of steel fiber cannot change the characteristic that the normal section of the corbel does not comply with the plane section assumption. From Figure 8b, it can be seen that the strain of steel-fiber-reinforced high-strength concrete corbel diagonal section concrete was not as regular as that of the normal section, which was related to the complex stress state inside the concrete diagonal strut.

## 3. Calculation Methods of National Codes

The truss theory, which equates steel bar and concrete as truss ties and struts, is an early mechanical model of concrete corbels. Based on the triangular truss model (see Figure 9), the former Ministry of Metallurgy Building Research Institute proposed a practical calculation formula for determining the cross-sectional size and bearing capacity of corbels by analyzing a large number of test results [27], which became the basis for the design of the corbels in the GB50010-2010 code. The ACI318-19 code, EC1992-1-1 code and CSA A23.3-19 code suggest determining the ultimate bearing capacity of corbels by establishing a strut-and-tie model. The strut-and-tie model is a lower-bound analysis method of plastic mechanics [28], which abstracts the corbel into a discrete truss model consisting of a longitudinal reinforcement tie subjected to tensile stress, concrete struts subjected to compressive stress and nodes connecting the tie and struts. The actual shape of the concrete struts is that of bottle-shaped struts, but it is usually idealized as prismatic struts [29], as shown in Figure 10. The specific calculation formulas for the national codes are shown in Table 6.

## 4. The Calculation Model of Shear Bearing Capacity of Corbels Based on Softened Strut-and-Tie Model (MSSTM)

### 4.1. Shear Mechanism

In the softened strut-and-tie model, there are usually three load paths to transmit diagonal compression, which are diagonal, horizontal and vertical mechanisms. The vertical stirrups in the corbels with small shear span ratio have a limited effect on the shear bearing capacity of the section. Therefore, the vertical mechanism is ignored in the corbel stress model. An incomplete, softened strut-and-tie model consisting of diagonal and horizontal mechanisms has been proposed for the corbel [30]. The diagonal mechanism is a steel-fiber concrete diagonal strut with an inclination angle of θ. The horizontal mechanism is composed of a horizontal tie and two gentle struts with the inclination angle of ${\theta^{\prime }}_{s}$, as shown in Figure 11. The inclination angle θ is defined as θ=arctan(jd/a), and the inclination angle ${\theta \prime }_{s}$ is defined as ${\theta \prime }_{s}=arctan(jd/2a)$, where jd is the lever arm. According to the linear bending theory, jd can be estimated as:(1)$$jd=h_{0}-\frac{kh_{0}}{3}$$
(2)$$k=\sqrt{{\left(n{\rho \prime }_{f}\right)}^{2}+2n{\rho \prime }_{f}}-n{\rho \prime }_{f}$$
(3)$${\rho \prime }_{f}=\frac{A_{s}+\Omega A_{s,h}}{bh_{0}}$$
where $h_{0}$ is the effective height of the corbels; n is the modular ratio of elasticity, that is, $n=E_{s}/E_{c}$; ${\rho \prime }_{f}$ is the ratio of bending tensile reinforcement, $A_{s}$ is the cross-sectional area of longitudinal reinforcement; $A_{s,h}$ is the cross-sectional area of horizontal stirrup; and Ω is the stirrup influence coefficient (Ω=0.2 in this paper).

The effective area of the diagonal strut $A_{str}$ is defined as:(4)$$A_{str}=a_{s}\times b=kh_{0}b$$
where $a_{s}$ is the width of the diagonal strut, and the definition of $a_{s}$ is different from the width of the diagonal strut $w_{s}$ in the strut-and-tie model.

The addition of steel fibers improves the tensile strength of concrete, making the mechanical performance of a steel-fiber-reinforced concrete corbel different from that of a reinforced concrete corbel. The horizontal tie is composed of two parts, the horizontal stirrups and the steel fibers, so [31]:(5)$$F_{h}=F_{s,h}+F_{sf,h}$$
(6)$$F_{s,h}=\eta_{1}A_{s,h}f_{s,h}$$
(7)$$F_{sf,h}=A_{sf,h}f_{sf,h}$$
where $F_{s,h}$, $F_{sf,h}$ are the tension of the horizontal stirrup tie and the horizontal steel-fiber tie, respectively; $A_{s,h}$, $A_{sf,h}$ are the cross-sectional areas of horizontal stirrups and horizontal steel-fiber ties, respectively. $\eta_{1}$ is the effective coefficient of the horizontal stirrups’ shear resistance. According to the test results, the horizontal stirrups did not fully yield when the specimen was damaged. According to Hwang et al. [30], it is roughly assumed that 75% of horizontal stirrups are fully utilized, thus $\eta_{1}=0.75$.

Steel fibers are randomly distributed in the concrete. Steel fibers are equivalent to an equal number of micro horizontal and vertical reinforcements [31]. Therefore, $A_{sf,h}$ is calculated as follows:(8)$$A_{sf,h}=n_{sf,h}A_{sf}$$
where $A_{sf}$ is the cross-sectional area of a single steel fiber; $n_{sf,h}$ is the number of equivalent horizontal steel fibers, which is calculated by the following formula:(9)$$n_{sf,h}=\eta_{2}\rho_{f}\frac{bh}{cos\theta }\frac{1}{A_{sf}}$$
where $\rho_{f}$ is the steel fiber volume fraction; $\eta_{2}$ is the equivalent coefficient, and the approximate value of $\eta_{2}$ is 0.41 [32]. Therefore, $A_{sf,h}=0.41\rho_{f}\frac{bh}{cos\theta }$.

### 4.2. Force Equilibrium

The internal force distribution mechanism of the softened strut-and-tie model is shown in Figure 12. According to the equilibrium condition, the resistances against the vertical shear and the horizontal shear are jointly borne by the diagonal and horizontal force transmission mechanisms, that is,
(10)$$V_{jv}=-Dsin\theta +F_{h}tan\theta$$
(11)$$V_{jh}=-Dcos\theta +F_{h}$$
where D is the compression force in the diagonal strut; $F_{h}$ is the tension force in the horizontal tie. The Equations (10) and (11) satisfy the relationship equation $V_{jv}/V_{jh}=tan\theta$.

It is assumed that the ratios of the vertical shear $V_{jv}$ allocated between the two mechanisms is [30]:(12)$$-Dsin\theta :\;F_{h}tan\theta =R_{d}:\;R_{h}$$
where $R_{d}$ and $R_{h}$ are, respectively, the ratios of the corbel shear resisted by the diagonal and horizontal mechanisms, which satisfy $R_{d}+R_{h}=1$. The values of $R_{d}$ and $R_{h}$ are calculated by the following equations [33]:(13)$$R_{h}=\gamma_{h}$$
(14)$$R_{d}=1-\gamma_{h}$$
(15)$$\gamma_{h}=\frac{2tan\theta -1}{3}$$
where $\gamma_{h}$ is the ratio of the horizontal tie to the horizontal shear without a vertical tie, and the value range of $\gamma_{h}$ is $0{\leq \gamma }_{h}\leq 1$. Two boundary conditions need to be met, namely, that the whole shear force is endured entirely by the horizontal mechanism when θ≥arctan⁡(2) and that the whole shear force is endured entirely by the diagonal mechanism when θ≤arctan⁡(1/2).

The failure of the specimen can be considered as being when the maximum compressive forces in the nodal zone $\sigma_{d,max}$ caused by the diagonal and flat struts in the resistance mechanisms reach the ultimate strength of the concrete; $\sigma_{d,max}$ can be expressed as:(16)$$\sigma_{d,max}=\frac{1}{A_{str}}\{D-\frac{cos({\theta -\theta \prime }_{s})}{cos{\theta \prime }_{s}}F_{h}\}$$

### 4.3. Constitutive Laws

According to Zhang and Hsu [34], the softening stress–strain relationship of cracked steel-fiber-reinforced concrete can be expressed as:(17)$$\left\{\begin{array}{cc} \sigma_{d}=-\zeta {f^{\prime }}_{c}\left[2\left(\frac{-\varepsilon_{d}}{\zeta \varepsilon_{0}}\right)-{\left(\frac{-\varepsilon_{d}}{\zeta \varepsilon_{0}}\right)}^{2}\right], & \frac{-\varepsilon_{d}}{\zeta \varepsilon_{0}}\leq 1\\ \sigma_{d}=-\zeta {f^{\prime }}_{c}\left[1-{\left(\frac{-\varepsilon_{d}/\zeta \varepsilon_{0}-1}{2/\zeta -1}\right)}^{2}\right], & \frac{-\varepsilon_{d}}{\zeta \varepsilon_{0}}>1 \end{array}\right.$$
(18)$$\left\{\begin{array}{c} \zeta_{1}=\frac{5.8}{\sqrt{{f^{\prime }}_{c}}}\;\frac{1}{\sqrt{1+400\varepsilon_{r}}}\;\leq \frac{0.9}{\sqrt{1+400\varepsilon_{r}}}\\ \;\zeta_{2}=\frac{5.8}{\sqrt{{f^{\prime }}_{c}}}\;\frac{1}{\sqrt{1+600\varepsilon_{r}}}\;\leq \frac{0.9}{\sqrt{1+600\varepsilon_{r}}}\; \end{array}\right.$$
where $\sigma_{d}$ is the principal stress of concrete in the *d*-direction; *ζ* is the softening coefficient of steel-fiber-reinforced concrete. Relevant studies have shown that the softening coefficients of ordinary-strength-grade steel-fiber-reinforced concrete and steel-fiber high-strength concrete are different [35,36], being valued according to either $\zeta_{1}$ or $\zeta_{2}$, respectively, in Equation (18); ${f^{\prime }}_{c}$ is the compressive strength of the concrete cylinder; $\varepsilon_{d}$ and $\varepsilon_{r}$ are the principal compressive strain and principal tensile strain of concrete, respectively. Peak strain $\varepsilon_{0}$ of steel-fiber-reinforced concrete can be calculated by the following equation:(19)$$\varepsilon_{0}=0.002+0.001\left(\frac{{f^{\prime }}_{c}-20}{80}\right),20\leq {f^{\prime }}_{c}\leq 100\;MPa$$

When the following equations are met, it can be determined that the corbel has reached its ultimate shear bearing capacity:(20)$$\sigma_{d,max}=-\zeta {f^{\prime }}_{c}$$
(21)$$\varepsilon_{d}=-\zeta \varepsilon_{0}$$

It is assumed that the stress–strain relationship of steel bars satisfies the complete elastic–plastic model:(22)$$\left\{\begin{array}{c} f_{s,h}=E_{s}\varepsilon_{s,h}\;\;,\;\varepsilon_{s,h}<\varepsilon_{y,h}\;\;\\ f_{s,h}=f_{y,h}\;\;\;\;,\;\varepsilon_{s,h}\geq \varepsilon_{y,h} \end{array}\right.$$
where $E_{s}$ is the elastic modulus of the steel bars; $f_{s,h}$ and $\varepsilon_{s,h}$ are the tensile stress and strain of the horizontal stirrups, respectively; $f_{y,h}$ and $\varepsilon_{y,h}$ are the yield stress and strain of the horizontal stirrups, respectively.

Then, Formula (6) can be changed to:(23)$$F_{s,h}=0.75A_{s,h}E_{s}\varepsilon_{s,h}$$

The stress–strain relationship of steel fibers can be expressed as:(24)$$f_{sf}=E_{sf}\varepsilon_{sf}$$
where $E_{sf}$ is the elastic modulus of steel fibers, the value of which is ${2\times 10}^{5}\;MPa$ [31]; $f_{sf}$ and $\varepsilon_{sf}$ are the tensile stress and strain of the steel fiber, respectively.

Formula (7) can be changed to:(25)$$F_{sf,h}=A_{sf,h}E_{sf}\varepsilon_{sf,h}$$

The steel fibers are mainly pulled out rather than broken when the specimens are destroyed, owing to fact that the tensile strength of the steel fibers is relatively high. The shear resistance of the steel fibers is limited by the bonding strength of steel-fiber concrete. Therefore, $f_{sf}$ needs to meet the following conditions:(26)$$A_{sf}f_{sf}\leq \lambda_{sf}A_{spf}\tau_{sf,max}$$
where $\tau_{sf,max}$ is the maximum bond strength between steel fiber and concrete, and that $\tau_{sf,max}=2.5f_{ct}$ [37], and $f_{ct}$ is the tensile strength of steel-fiber concrete; $\lambda_{sf}$ is the influencing factor for different types of steel fibers, with values of 0.5, 0.75 and 1.0 for long straight, wavy and curved steel fibers, respectively [38] (the value for the experiment is 1.0); $A_{spf}$ is the surface area of the steel fiber, and $A_{spf}=\pi d_{sf}l_{sf0}$, in which $d_{sf}$ is the diameter of steel fiber and $l_{sf0}$ is the effective anchorage length of steel fiber, taken as $l_{sf0}=l_{sf}/4$, and $l_{sf}$ is the length of steel fiber.

Formula (26) can be changed to:(27)$$f_{sf}\leq \lambda_{sf}\left(\frac{l_{sf}}{d_{sf}}\right)\tau_{sf,max}$$

### 4.4. Strain Compatibility

The analysis of two-dimensional membrane elements should strictly satisfy the Mohr circle strain compatibility condition, namely:(28)$$\varepsilon_{r}+\varepsilon_{d}=\varepsilon_{h}+\varepsilon_{v}$$
where $\varepsilon_{d}$ and $\varepsilon_{r}$ are the principal compressive strain and principal tensile strain of steel-fiber-reinforced concrete, respectively; $\varepsilon_{h}$ and $\varepsilon_{v}$ are the average strains in the horizontal and vertical directions of steel-fiber-reinforced concrete, respectively.

Then, $\varepsilon_{h}$ is solved by combining Equations (23) and (25), satisfying $\varepsilon_{s,h}=\varepsilon_{sf,h}=\varepsilon_{h}$ [39], and thus obtaining:(29)$$\varepsilon_{h}=\frac{F_{h}}{0.75A_{s,h}E_{s}+A_{sf,h}E_{sf}}$$

Since there is no detailed description of the vertical stirrup in the corbel, there is no vertical tie that limits the development of horizontal cracks. The value of $\varepsilon_{v}$ is 0.002 for the corbel with a/jd>1/2 and it is 0 for the corbel with a/jd≤1/2 [30].

### 4.5. Solution Process

The shear bearing capacity of corbels $V_{jv}$ is solved by the equilibrium equations, constitutive laws and compatibility equation above, and the solution process is shown in Figure 13. The detailed calculation steps are as follows:

(1)Input the basic parameters of the corbel: a, b, $h_{0}$, $A_{s}$, $A_{sh}$, $A_{str}$, ${f^{\prime }}_{c}$, etc.;(2)$R_{d}$ and $R_{h}$ are calculated through Equations (13) to (15);(3)Select the initial $V_{jv}$, and solve *D* and $F_{h}$ using Equations (10) and (12);(4)Calculate $\sigma_{d,max}$ and ${{\varepsilon_{h}}_{}}_{}$ through Equations (16) and (29), and take values of $\varepsilon_{v}$ reasonably based on the size of the corbels;(5)Given $\varepsilon_{d}$ reasonably, ${\varepsilon_{r}}_{}$ is solved using Equation (28), and then the softening coefficient *ζ* is solved using Equation (18);(6)$\varepsilon_{0}$ is calculated using Formula (19), and then the value $\sigma_{d}$ corresponding to a given $\varepsilon_{d}$ is calculated by the steel bar constitutive Equation (17);(7)Compare the values of $\left|\sigma_{d,max}\right|$ and $\left|\sigma_{d}\right|$, and if $\left|\sigma_{d,max}\right|\geq \left|\sigma_{d}\right|$, repeat steps (3) to (6) until $\;\left|\sigma_{d,max}\right|\geq \left|\sigma_{d}\right|$;(8)After satisfying step (7), compare ${|\varepsilon }_{d}|$ and ${|\zeta \varepsilon }_{0}|$. If $\;{|\varepsilon }_{d}|\geq {|\zeta \varepsilon }_{0}|$, repeat steps (5) to (7) until ${|\varepsilon }_{d}|\geq {|\zeta \varepsilon }_{0}|$, at which point the solving is completed. At this point, the maximum stress in the node zone $\sigma_{d,max}$ has reached the strength limit of the concrete, and the $V_{jv}$ is the bearing capacity value of the corbel.

## 5. Simplification of the Calculation Model for the Shear Bearing Capacity of the Corbels (SMSSTM)

Due to the iterative process of the softened strut-and-tie model for a steel-fiber-reinforced concrete corbel being complicated, the model was simplified to fully describe the shear mechanism of the corbel while reducing the variables.

Hwang et al. [30] pointed out in a study on the shear strength analysis of corbels that there is a bilinear relationship between the shear strength of corbels and horizontal reinforcement. The compressive strength of concrete seems to set an upper limit value for the amount of horizontal reinforcement. Below this value, the compressive strength of concrete increases with the increase in the horizontal reinforcement. At this equilibrium limit, the shear strength of the corbel will also reach its extreme value when the horizontal reinforcement yields. When the amount of horizontal reinforcement exceeds the limit value, the horizontal reinforcement exceeding the equilibrium amount only plays a role in delaying the softening effect of concrete, and it has little effect on preventing the reduction of concrete compressive strength, so its beneficial effect on shear strength can be ignored. The linear relationship between shear strength and reinforcement shows that the shear strength prediction of corbels that do not reach the equilibrium reinforcement can be obtained by linear interpolation.

The beneficial effect of horizontal ties in improving the shear strength of corbels can be understood as follows: when the corbels are reinforced with horizontal stirrups, diagonal compression will be borne by additional load paths in addition to the diagonal struts, which will activate more concrete to participate in shear, thereby improving the shear strength of the corbels. The beneficial effect of horizontal ties on shear strength can be expressed by the index K [40]:(30)$$K=\frac{C_{d}}{{-\sigma }_{d,max}\times A_{str}}=\frac{-D+\frac{F_{h}}{cos\theta }}{-D+\frac{cos\;({\theta -\theta \prime }_{s})}{cos{\theta \prime }_{s}}F_{h}}\geq 1$$

Correspondingly, the nominal diagonal compressive strength of the corbel can be expressed as [41]:(31)$$C_{d,n}=K\zeta {f^{\prime }}_{c}A_{str}$$
where $C_{d,n}$ is the nominal diagonal compressive strength of corbels; K is the strut-and-tie index.

### 5.1. Approximation of Softening Effect

The softening coefficient of concrete is directly related to the value of the principal tensile strain $\varepsilon_{r}$. Vecchio et al. [42] suggested that the value of $\varepsilon_{r}$ should be limited to the strain level at which the reinforcement yields through a crack. In order to simplify the calculation, the values of $\varepsilon_{h}$ and $\varepsilon_{v}$ are both 0.002, and $\varepsilon_{d}$ can be approximately taken as −0.001 [24], so the value of $\varepsilon_{r}$ calculated using Equation (28) is 0.005 (0.002 + 0.002 + 0.001). Therefore, the softening coefficient ζ can be expressed as:(32)$$\left\{\begin{array}{c} \zeta_{1}=3.35/\sqrt{{f^{\prime }}_{c}}\leq 0.52\\ \zeta_{2}=2.90/\sqrt{{f^{\prime }}_{c}}\leq 0.45 \end{array}\right.$$

### 5.2. Strut-and-Tie Index K

For discontinuous region components subjected to diagonal compression, there are usually four possible combinations of resistance mechanisms, namely, the diagonal mechanism, the diagonal plus horizontal mechanisms, the diagonal plus vertical mechanisms, and complete mechanisms. For corbels, only two incomplete load-bearing mechanisms, diagonal mechanism and diagonal plus horizontal mechanisms, could be developed in shear analysis. The determination process of the strut-and-tie index K for each combination is as follows.

If there is no horizontal tie reinforcement in the corbel, the diagonal compression $C_{d}$ is only provided by the diagonal mechanism, as shown in Figure 14. Then, through Equation (30), we can obtain:(33)$$K_{d}=(-D)/(-D)=1$$
where $K_{d}$ is the diagonal strut index.

If the flow of forces at nodes is realized by the strut-and-tie model composed of the diagonal and horizontal mechanisms shown in Figure 12, more flat struts will be formed, and the internal force flows deviate from the direction of $C_{d}$ [24]. When there are sufficient horizontal reinforcements, the concrete strut would reach its compressive strength when the horizontal tie is maintained within the elastic range, and the index at this time is:(34)$$\bar{K_{h}}=\frac{(1-\gamma_{h})+\gamma_{h}}{(1-\gamma_{h})+\gamma_{h}\left(1-\frac{{sin}^{2}\theta }{2}\right)}\geq 1$$

Formula (34) above can be further simplified to:(35)$$\bar{K_{h}}\approx \frac{1}{1-0.2(\gamma_{h}+{\gamma^{2}}_{h})}$$
where $\bar{K_{h}}$ is the horizontal tie index with sufficient horizontal reinforcement. At this time, the balanced amount of the horizontal tie force $\bar{F_{h}}$ is:(36)$$\bar{F_{h}}=\gamma_{h}\times (\bar{K_{h}}\zeta {f^{\prime }}_{c}A_{str})\times cos\theta$$

In the case of insufficient reinforcement, the horizontal tie index $K_{h}$ after linear interpolation is approximately:(37)$$K_{h}=1+(\bar{K_{h}}-1)\times F_{yh}/\bar{F_{h}}\leq \bar{K_{h}}$$
(38)$$F_{yh}=0.75A_{s,h}f_{y,h}+A_{sf,h}f_{sf,yh}=\left(0.75A_{s,h}E_{s}+A_{sf,h}E_{sf}\right)\varepsilon_{yh}$$
where $F_{yh}$ is the yield strength of the horizontal tie; $f_{y,h}$ and $f_{sf,yh}$ are the yield stresses of the horizontal stirrup tie and the horizontal steel-fiber tie, respectively; $\varepsilon_{yh}$ is the yield strain of the horizontal tie.

Therefore, for corbels with sufficient horizontal reinforcement, the value of the strut-and-tie index $\bar{K}$ is:(39)$$\bar{K}=\bar{K_{h}}$$

Similarly, in the case of insufficient horizontal reinforcement, the value of K can be estimated approximately as:(40)$$K=K_{h}$$

## 6. Example Verification

The test results for the six steel-fiber-reinforced high-strength concrete corbels in this paper were calculated by using the abovementioned national codes and the established calculation model for the shear bearing capacity of corbels, and the test values and calculated values were compared and analyzed. The results are shown in Table 7. From the table, we can see that:(1)The mean values of the ratios of the test values to the calculated values of the GB50010-2010 code, the ACI318-19 code, the EN 1992-1-1 code and the CSA A23.3-19 code were 1.564, 1.373, 1.533 and 1.153, respectively, and the variances were 0.131, 0.025, 0.046 and 0.011, respectively. It can be seen that all the calculation results were lower than the experimental values, and the results were conservative, which indicates that the various national codes are applicable to the calculation of the shear capacity of steel-fiber-reinforced concrete corbels. However, the calculation results of the GB50010-2010 code were relatively discrete, while the variances of the calculation results of the ACI318-19 code, the EN 1992-1-1 code and the CSA A23.3-19 code based on the strut-and-tie model were relatively small.(2)The mean value of the ratio of the test value to the calculated value of the established shear bearing capacity calculation model was 0.965, and the variance was 0.016. The mean value of the ratio between the experimental value and calculation value of the simplified model was 1.014, and the variance was 0.008. The calculated results were in good agreement with the experimental values. The established model for calculating the shear bearing capacity of the corbel has a clear mechanical concept, which can be used for the calculation and prediction of the shear bearing capacity of steel-fiber-reinforced high-strength concrete corbels.

## 7. Conclusions

The experimental studies showed that the steel-fiber-reinforced high-strength concrete corbels with shear span ratio from 0.2 to 0.5 mainly exhibit two typical failure modes: shear failure and diagonal compression failure. With an increase in shear span ratio, the bearing capacity of the corbel specimens decreases gradually, and the failure mode of specimens transitions from shear failure to diagonal compression failure. At the same time, with the same shear span ratio and the addition of steel fibers, the bearing capacity of the corbels increases with the increase in the steel fiber volume fraction.Steel fiber can improve the ductility of reinforced concrete corbels, but it has little effect on the failure mode of the diagonal section. The addition of steel fiber cannot change the characteristic that the normal section of the corbel does not comply with the plane section assumption.The shear mechanism of steel-fiber-reinforced high-strength concrete corbels can be described by the softened strut-and-tie model composed of diagonal struts and horizontal resistance mechanisms. In the analysis of shear capacity, randomly distributed steel fibers in concrete can be equivalent to an equal number of horizontal and vertical reinforcements, making the contribution of steel fiber dimensions to the bearing capacity of corbels more clear.The shear bearing capacity of steel-fiber-reinforced high-strength concrete corbels was calculated by using the national codes and the established corbel shear bearing capacity calculation model in this paper. The results showed that the calculation results of the national codes are lower than the experimental values, and the results are conservative. The national codes can be used for the shear bearing capacity calculation of steel-fiber-reinforced high-strength concrete corbels. However, the calculation results of the Chinese code (GB50010-2010) were more discrete, while the variances of the calculation results of the American code (ACI318-19), European code (EN 1992-1-1) and Canadian code (CSA A23.3-19), which are based on the strut-and-tie model, were relatively small. The theoretical calculation of the shear capacity calculation model for the corbels established in this paper was close to the experimental results. Moreover, the model has a clear mechanical concept considering the tensile properties of steel-fiber-reinforced concrete and the influence of horizontal stirrups, which can reasonably reflect the stress mechanism of the corbels and provide a theoretical reference for improving the design of steel-fiber-reinforced high-strength concrete corbels.

## Figures and Tables

**Figure 1 materials-16-04907-f001:**
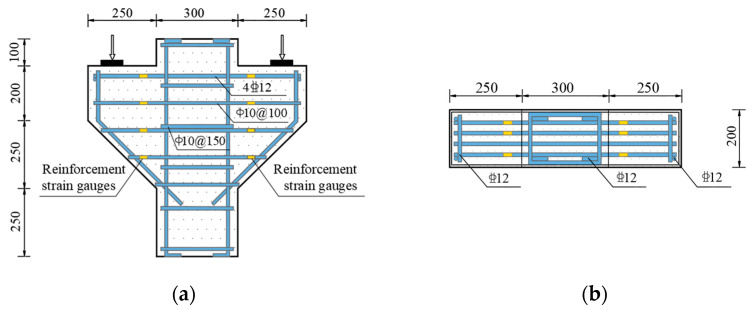
Arrangement of reinforcement and reinforcement strain measuring points. (**a**) Front view. (**b**) Bird’s-eye view.

**Figure 2 materials-16-04907-f002:**
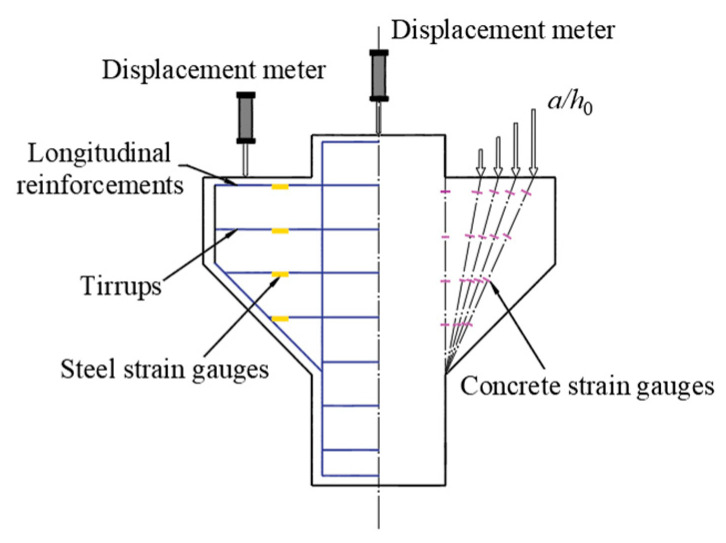
The layout of strain measuring points and displacement meters.

**Figure 3 materials-16-04907-f003:**
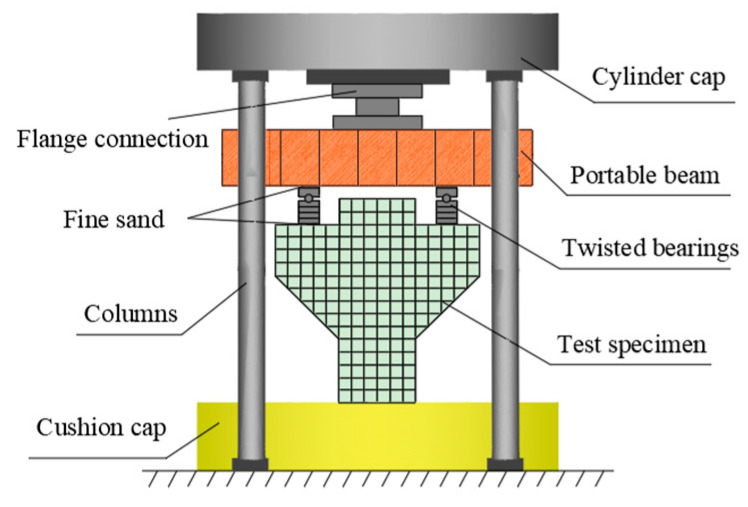
Test loading device.

**Figure 4 materials-16-04907-f004:**
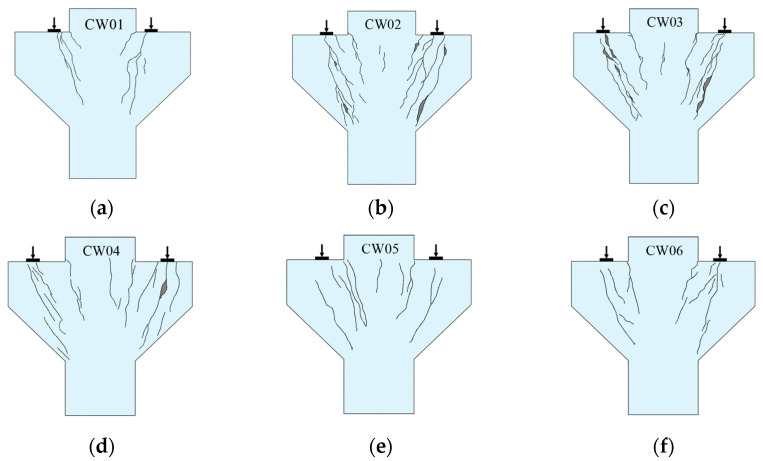
Crack morphology of the specimens. (**a**) CW01; (**b**) CW02; (**c**) CW03; (**d**) CW04; (**e**) CW05; (**f**) CW06.

**Figure 5 materials-16-04907-f005:**
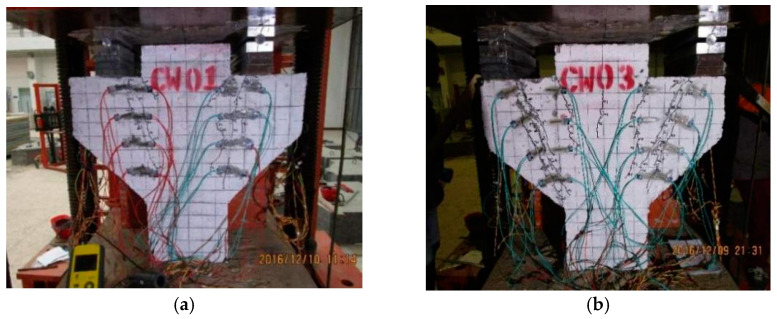
Typical specimen failure mode. (**a**) CW01 (shear failure). (**b**) CW03 (diagonal compression failure).

**Figure 6 materials-16-04907-f006:**
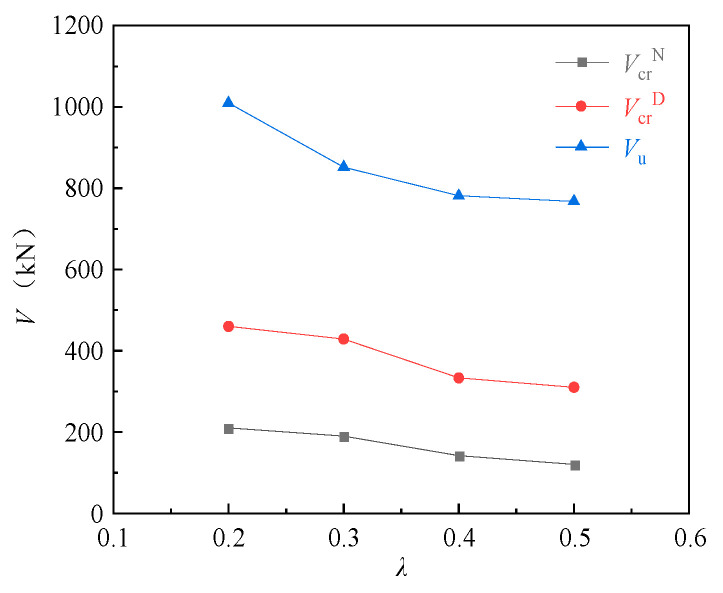
Relationship curve for load to shear span ratio.

**Figure 7 materials-16-04907-f007:**
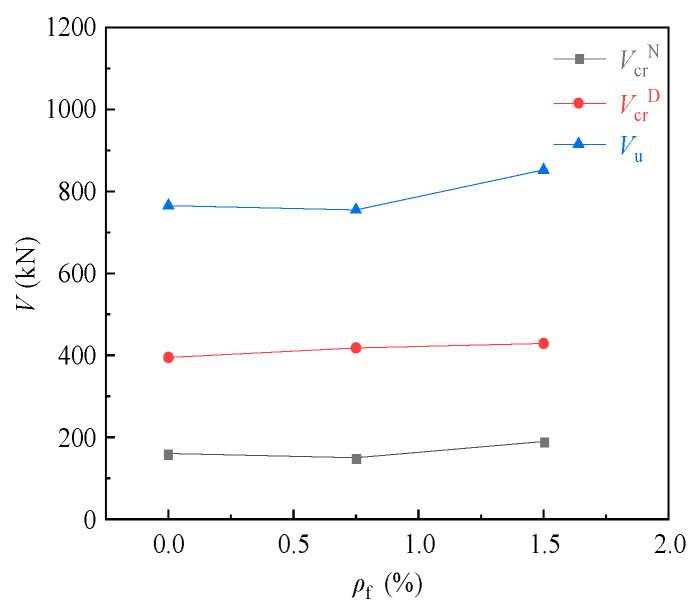
Relationship curve for load to steel fiber volume fraction.

**Figure 8 materials-16-04907-f008:**
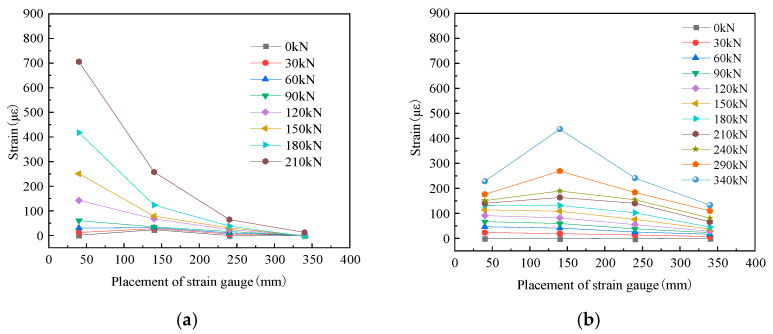
Concrete strain distribution of specimen CW02. (**a**) Normal section strain. (**b**) Diagonal section strain.

**Figure 9 materials-16-04907-f009:**
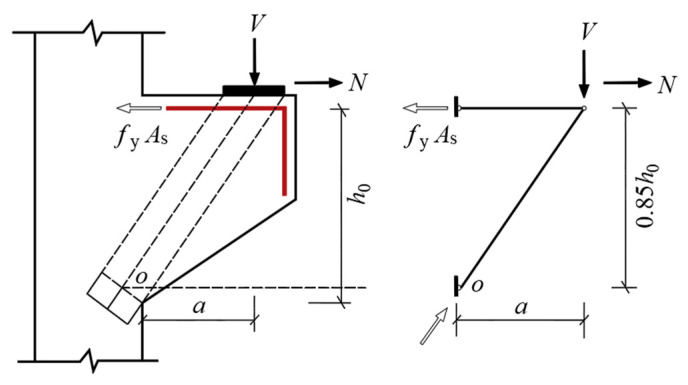
Triangle truss model.

**Figure 10 materials-16-04907-f010:**
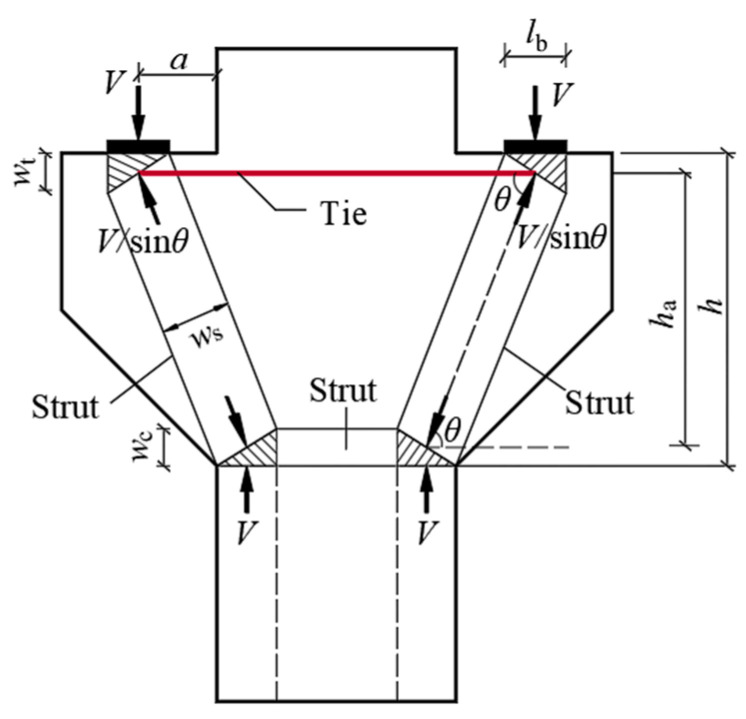
The strut-and-tie model of corbel.

**Figure 11 materials-16-04907-f011:**
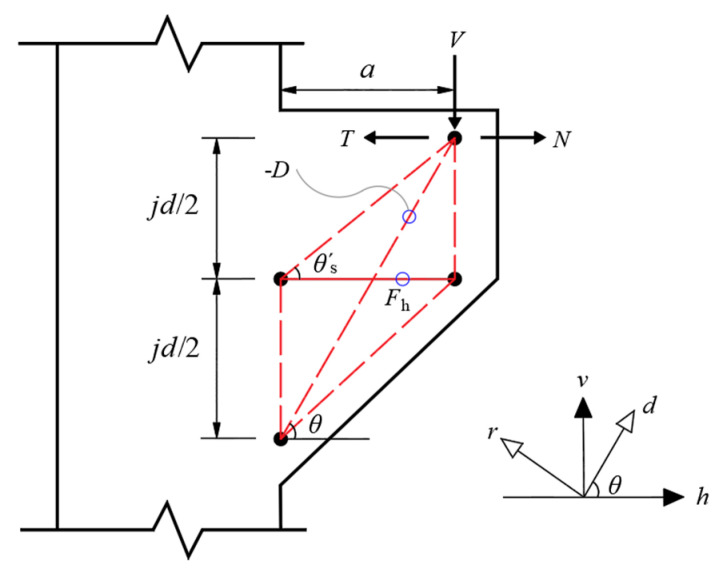
The softened strut-and-tie model of corbel.

**Figure 12 materials-16-04907-f012:**
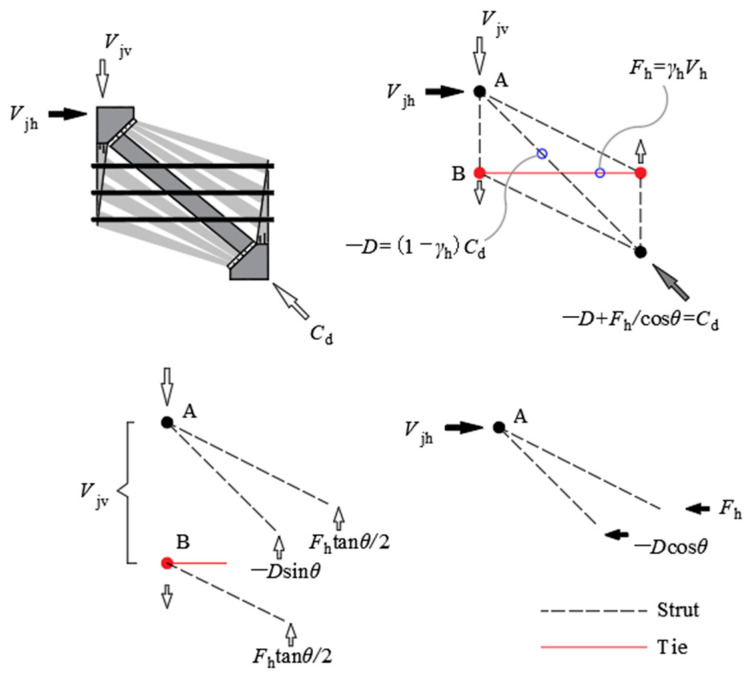
Internal force distribution of the softened strut-and-tie model for corbel.

**Figure 13 materials-16-04907-f013:**
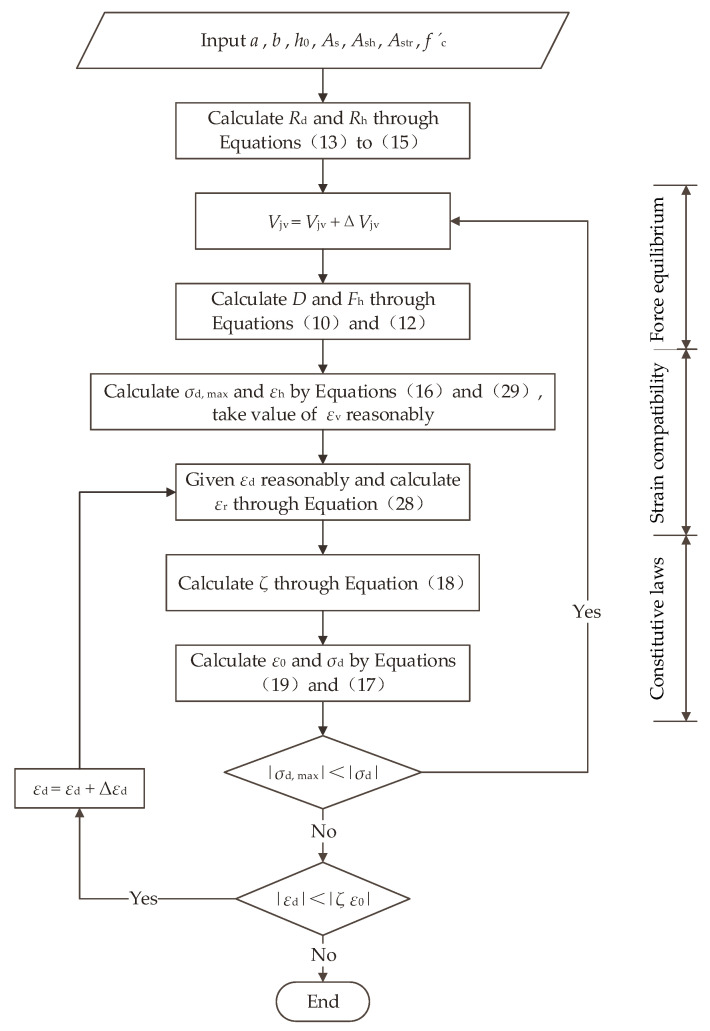
Solution flowchart.

**Figure 14 materials-16-04907-f014:**
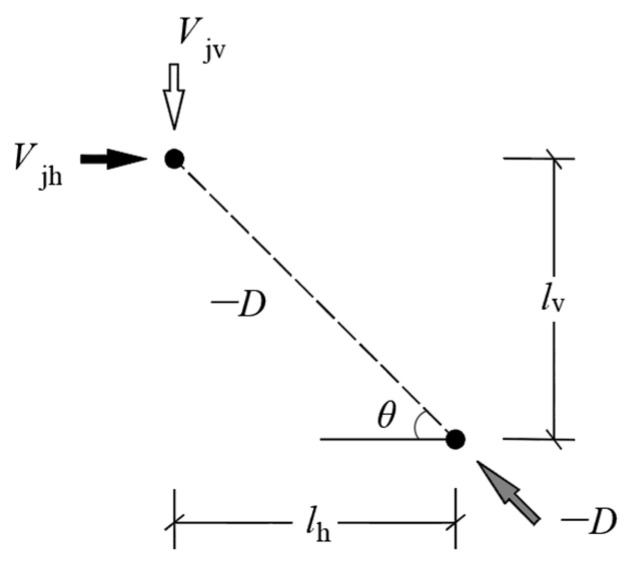
Diagonal mechanism.

**Table 1 materials-16-04907-t001:** Design parameters of specimens.

Specimen Number	λ	Longitudinal Reinforcement	$\rho_{s}$ (%)	Stirrup	$\rho_{sh}$ (%)	$\rho_{f}$ (%)
CW01	0.2	4C12	0.55	A10@100	0.78	1.50
CW02	0.3	4C12	0.55	A10@100	0.78	1.50
CW03	0.4	4C12	0.55	A10@100	0.78	1.50
CW04	0.5	4C12	0.55	A10@100	0.78	1.50
CW05	0.3	4C12	0.55	A10@100	0.78	0.75
CW06	0.3	4C12	0.55	A10@100	0.78	0

λ is the shear-span ratio; $\rho_{s}$ is the longitudinal reinforcement ratio; $\rho_{sh}$ is the stirrup ratio; $\rho_{f}$ is the steel fiber volume fraction.

**Table 2 materials-16-04907-t002:** Mix proportion of SFHSC.

$\rho_{f}$ (%)	Water (kg)	Cement (kg)	Sand (kg)	Pebble (kg)	Water Reducer (kg)	Steel Fiber (kg)	Gross Mass (kg)
0	61.0	203.4	249.0	483.4	3.05	0	1000
0.75	65.1	216.9	285.7	428.8	3.26	23.80	1000
1.5	69.9	233.1	277.3	415.6	3.50	47.59	1000

**Table 3 materials-16-04907-t003:** Mechanical properties of SFHSC.

$\rho_{f}$ (%)	$f_{cu}$ (MPa)	$f_{c}$ (MPa)	$f_{t}$ (MPa)	$E_{c}$ (MPa)
0	73.2	55.9	4.0	38.7
0.75	69.8	51.7	3.3	37.6
1.5	72.6	49.8	3.6	37.2

$f_{cu}$ is the cube compressive strength of concrete; $f_{c}$ is the prism compressive strength of concrete; $f_{t}$ is the tensile strength of concrete; $E_{c}$ is the elastic modulus of concrete.

**Table 4 materials-16-04907-t004:** Mechanical properties of reinforcement.

Reinforcement Type	$A_{s}$ $\;({mm}^{2})$	$f_{y}$ (MPa)	$E_{s}$ (GPa)	$f_{u}$ (MPa)	$\varepsilon_{y}(10^{-6})$
A10	78.5	333.7	195.95	535.8	1703
C12	113.1	425.2	172.60	541.4	2463

$A_{s}$ is the cross-sectional area of reinforcement; $f_{y}$ is the yield strength of reinforcement; $E_{s}$ is the elastic modulus of reinforcement; $f_{u}$ is the ultimate strength of reinforcement; $\varepsilon_{y}$ is the yield strain of reinforcement.

**Table 5 materials-16-04907-t005:** Test results and failure modes of specimens.

Specimen Number	λ	$\rho_{f}$ (%)	$V_{cr}^{N}$ (kN)	$V_{cr}^{D}$ (kN)	$V_{u}$ (kN)	$V_{cr}^{N}/V_{u}$ (%)	$V_{cr}^{D}/V_{u}$ (%)	Failure Mode
CW01	0.2	1.50	210.0	460.0	1009.5	20.8	45.6	Shear failure
CW02	0.3	1.50	190.0	429.0	852.0	22.3	50.4	Diagonal compression
CW03	0.4	1.50	142.0	333.0	781.5	18.2	42.6	Diagonal compression
CW04	0.5	1.50	120.0	310.0	767.5	15.6	40.4	Diagonal compression
CW05	0.3	0.75	150.0	418.0	755.0	19.9	55.4	Diagonal compression
CW06	0.3	0	160.0	395.0	765.0	20.9	51.6	Diagonal compression

Note: The loads in the table are the loads on one side of the corbels.

**Table 6 materials-16-04907-t006:** Calculation models of shear bearing capacity for corbels in national codes.

Formula Source	Calculation Formulas	Parameters
Chinese code [12] (GB50010-2010)	$V=\frac{0.85f_{y}h_{0}}{a}\left(A_{s}-1.2\frac{N}{f_{y}}\right)$	V and N are the vertical load and the horizontal load acting on the top of the corbel, respectively; $A_{s}$ is the total area of longitudinal tensile reinforcements; $f_{y}$ is the yield strength of longitudinal tensile reinforcement; $h_{0}$ is the effective depth of corbel section; a is the horizontal distance between vertical load and the edge of the corbel lower column.
American code [13] (ACI318-19)	$V_{ACI}=0.85\beta_{c}\beta_{s}{f^{\prime }}_{c}bw_{s}sin\theta$ $w_{s}=w_{t}cos\theta +l_{b}sin\theta$ $\theta =\tan^{-1}\left(h_{a}/a\right)$ $h_{a}=h-w_{t}/2-w_{c}/2$	$\beta_{c}$ is the strut and node confinement modification factor; $\beta_{s}$ is the strut effective coefficient; ${f^{\prime }}_{c}$ is the compressive strength of concrete cylinders; b is the section width of the corbel; $w_{s}$ is the width of strut; θ is the angle between strut and horizontal direction; $h_{a}$ is the vertical distance between the top node center and the bottom node center; $l_{b}$ is the width of the loading plate; $w_{t}$$\;\mathrm{and}\;w_{c}$ are, respectively, the height of the node at the loading area and at the supporting area, and considering the simplicity of the calculation, the values of the two are equal ($w_{c}=w_{t}$).
European code [14] (EN 1992-1-1)	$V_{EN2}=\sigma bw_{s}sin\theta$ $\nu =1-f_{ck}/250$ $f_{cd}=\alpha_{cc}f_{ck}/\gamma_{c}$	σ is the compressive stress in the concrete, the values of which are two cases: ① $\sigma =f_{cd}$ (when there is no transverse tensile stress or compressive stress); ② $\sigma =0.6vf_{cd}$ (when there is transverse tensile stress and cracking is allowed). ν is the strength reduction factor for concrete cracked in shear; $f_{cd}$ is the design value of concrete compressive strength; $\alpha_{cc}$ is the coefficient taking account of long-term effects on the compressive strength and of unfavorable effects resulting from the way the load is applied; $\gamma_{c}$ is the partial safety factor for concrete; $f_{ck}$ is the characteristic compressive cylinder strength of concrete at 28 days.
Canadian code [15] (CSA A23.3-19)	$V_{CSA}=\Phi_{c}f_{cu}bw_{s}sin\theta$ $f_{cu}=\frac{{f^{\prime }}_{c}}{\left(0.8+170\varepsilon_{1}\right)}\leq 0.85{f^{\prime }}_{c}$ $\varepsilon_{1}=\varepsilon_{s}+\left(\varepsilon_{s}+0.002\right){cot}^{2}{ϴ}_{s}$	$\Phi_{c}$ is the resistance factor for concrete, taken as 1; $f_{cu}$ is the limiting compressive stress in concrete strut; ${ϴ}_{s}$ is the smallest angle between compressive strut and adjoining tensile ties; $\varepsilon_{1}$ is the principal tensile strain in cracked concrete due to factored loads; $\varepsilon_{s}$$\;\mathrm{is}\;\mathrm{the}\;\mathrm{tensile}\;\mathrm{strain}\;\mathrm{corresponding}\;\mathrm{to}\;\mathrm{the}\;\mathrm{tensile}\;\mathrm{tie}\;\mathrm{with}\;\mathrm{the}\;\mathrm{inclination}\;\mathrm{angle}\;\mathrm{of}\;{ϴ}_{s}$ to concrete strut, which ranges from 0.0012 to 0.0038; particularly, when the specified yield strength of the steel bar does not exceed 400 MPa, $f_{cu}=\frac{{f^{\prime }}_{c}}{\left(1.14+0.68{cot}^{2}{ϴ}_{s}\right)}\leq 0.85{f^{\prime }}_{c}$.

**Table 7 materials-16-04907-t007:** Comparison of experimental results and calculation results for shear bearing capacity.

Specimen Number	$V_{test}$ (kN)	$V_{n}$ (kN)						$V_{test}/V_{n}$					
		GB10	ACI	EC2	CSA	MSSTM	SMSSTM	GB10	ACI	EN2	CSA	MSSTM	SMSSTM
CW01	1009.5	858	591	515	740	1077	849	1.177	1.709	1.961	1.365	0.937	1.190
CW02	852.0	572	605	528	733	968	828	1.490	1.407	1.615	1.162	0.880	1.029
CW03	781.5	429	607	529	703	859	801	1.822	1.287	1.477	1.112	0.910	0.976
CW04	767.5	343	598	521	655	630	738	2.236	1.284	1.473	1.171	1.218	1.039
CW05	755.0	572	582	548	705	905	812	1.320	1.297	1.378	1.071	0.834	0.930
CW06	765.0	572	610	592	739	756	831	1.337	1.253	1.292	1.035	1.012	0.921
Mean								1.564	1.373	1.533	1.153	0.965	1.014
Variance								0.131	0.025	0.046	0.011	0.016	0.008

Note: $V_{test}$ is the test value; $V_{n}$ is the calculated value.

## Data Availability

Data are contained within the article.

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
