# Peer review of "Analysis of Shear Model for Steel-Fiber-Reinforced High-Strength Concrete Corbels with Welded-Anchorage Longitudinal Reinforcement"

_materials, 2023, doi:10.3390/ma16144907_

Round 1

Reviewer 1 Report

Title: Analysis of Shear Model for Steel Fiber Reinforced High-Strength Concrete Corbels with Welded Anchorage Longitudinal Reinforcement

General observation:

1.     This article focused on establishing a simplified calculation method for analyzing the shear bearing capacity of steel fiber reinforced high-strength concrete (SFHSC) corbels with steel fiber. Two variables have been selected for the experimental program: shear span ratio of 0.2, 0.3, 0.4 and 0.5, and steel fiber volume of 0%, 0.75% and 1.5%. The corbel specimens were loaded until failure, and the test results are presented elaborately. The calculation method to analyze the shear model of SFHSC is very detailed and has been presented appropriately. However, the reasons behind the experimental results are unclear from the manuscript. It would be better if the explanation for why the load and shear bearing capacity of SFHSC corbels decreased for 0.75% fiber and increased for 1.5% fiber was pointed out. Also, please include any relevant picture showing the difference in crack number/crack width of 0.75% and 1.5% steel fiber reinforced corbel.

2.     Please mention the mechanical properties of the steel fiber used in this study.

3.     The authors have performed load tests and several mechanical tests to investigate the behavior of SFHSC corbels. It would be better if the authors mentioned the test standard followed to perform these tests and the age of the specimens during the test.

4.     Each of the corbel specimens was prepared with 16 concrete strain gauges. However, the result section does not contain the contribution of these strain gauges. Please include their test data.

5.     This paper contains several equations containing constants. Please specify if those equations are independent of the unit of the parameters. If not, please mention their unit.

Specific Observation:

1.     Page 3, Line 140: Why did the authors choose the 32 mm length of the steel fibers? Please explain.  

2.     Page 3, Line 143: The authors have mentioned that they prepared concrete cubes and prism test blocks to determine the compressive strength, axial compressive strength, splitting tensile strength, and elastic modulus of SFHSC. Please mention the size of the specimens and which type of specimens was used to determine which property.

3.     How many corbel specimens are tested in a group? If there is more than one specimen in a group, are the presented results an average of those specimens? If only one specimen exists in a group, how do the authors justify the experimental results?

4.     Page 10, Line 303: Please mention the value of Ω.

5.     Page 11, Line 307: The statement “The addition of steel fibers improves the tensile strength of concrete” does not support Table 3. Please revise.

6.     Page 15, Line 420-426: Please rewrite this sentence. The sentence is too long.

7.     Page 18: Please include the influence of fiber in the conclusion section.

8.     Please use abbreviations consistently, such as SFHSC.

Some of the sentences are too long or complex. Please revise. 

Reviewer 2 Report

An article entitled »Analysis of shear model for steel fiber reinforced high-strength concrete corbels with welded anchorage longitudinal reinforcement« is very interesting for the field of construction. The authors have done a very high-quality research.

The summary is written comprehensibly. Authors should add the limitation of the research and the main objective of the research at the end of the introduction.

The experimental part is self-explanatory. The pictures are clear and well described.

Authors should use the term formulas instead of the equestions.

The equations are clear and well described, giving the article scientific weight.

I suggest that before the conclusion they add a discussion, where they should describe the set goal, expectations, limitations of the research, problems and where they see the research in the future and make a comparison with some other comparison that has already been done.

Literature is too limited to the domain of authors. Let's add more works that were done elsewhere in the world. A lot of this has already been done.

Final opinion: the article is interesting. We carried out similar research ourselves, where we reinforced and tested concrete beams with different materials, fibers, slats, hemp, coatings...

In any case, it is necessary to pay a lot of attention to this kind of research, because the consequences of cracks can be catastrophic.

It might make sense to add fracture testing in the future to see when and why the material gives way under load.
